# Auger-spectroscopy in quantum Hall edge channels and the missing energy problem

T. Krähenmann [1,5], S.G. Fischer[2,3], M. Röösli [1], T. Ihn[1], C. Reichl[1], W. Wegscheider[1], K. Ensslin [1], Y. Gefen[2] & Yigal Meir[3,4]

Quantum Hall edge channels offer an efficient and controllable platform to study quantum transport in one dimension. Such channels are a prospective tool for the efficient transfer of quantum information at the nanoscale, and play a vital role in exposing intriguing physics. Electric current along the edge carries energy and heat leading to inelastic scattering, which may impede coherent transport. Several experiments attempting to probe the concomitant energy redistribution along the edge reported energy loss via unknown mechanisms of inelastic scattering. Here we employ quantum dots to inject and extract electrons at specific energies, to spectrally analyse inelastic scattering inside quantum Hall edge channels. We show that the missing energy puzzle could be untangled by incorporating non-local Auger-like processes, in which energy is redistributed between spatially separate parts of the sample. Our theoretical analysis, accounting for the experimental results, challenges common-wisdom analyses which ignore such non-local decay channels.

[1] Solid State Physics Laboratory, ETH Zürich, CH-8093 Zürich, Switzerland. [2] Department of Condensed Matter Physics, Weizmann Institute of Science, Rehovot 76100, Israel. [3] Department of Physics, Ben-Gurion University of the Negev, Beer-Sheva 84105, Israel. [4] The Ilse Katz Institute for Nanoscale Science and Technology, Ben-Gurion University of the Negev, Beer-Sheva 84105, Israel. [5] Present address: QuTech and Kavli Institute of Nanoscience, Delft University of Technology, Delft 2628CJ, the Netherlands. Correspondence and requests for materials should be addressed to T.K. (email: tobiaskr@phys.ethz.ch)

The concept of quantum Hall edge channels is well supported by numerous experiments[1]. Directed transport in these channels serves as a platform to realise electronic counterparts of optical interferometers in mesoscopic devices[2–6]. The propagation of charge modes[7,8] as well as neutral modes[9–12] in quantum Hall edge channels have been investigated. Relaxation of non-equilibrium electrons between edge channels and possible coupling to the bulk of the sample, however, is not satisfactorily understood. In fact, several experiments indicated the existence of another, so far unknown channel for energy loss.

In a first experiment[8] addressing the equilibration of two edge channels in the integer quantum Hall regime, a non-equilibrium distribution has been injected into the outermost channel via a quantum point contact. Increasing the propagation distance, the channel has been probed by a quantum dot (QD), to record the gradual equilibration of the initial distribution. In the course of equilibration, a significant amount of energy is lost to degrees of freedom that are not controlled in the setup[13–15]. In a follow-up experiment at the same filling factor[16], the modes of the outer edge channel have been excited at specific energies by a radio frequency circuit, to be probed by an Ohmic contact downstream from the point of injection. That setup measured the dispersion of one of the ensuing eigenmodes, which showed that the chiral channels were dissipative. Also in that case, the concomitant energy loss remained unaccounted for.

Here, we demonstrate that, surprisingly, significant energy redistribution occurs also between sample components that are spatially well-separated. To this end, we use a QD to energy-selectively emit electrons from a biased Source contact into a quantum Hall edge at integer filling factor. This edge is subsequently probed by a second QD which serves as an energy-resolved detector. In the ensuing emitter-detector energy landscape currents are measured at detection energies that exceed emission energies, which ostensibly indicates a violation of energy conservation. The apparent contradiction can only be resolved by considering processes in which recombination energy is transferred from the Source contact to the edge channel probed by the Detector QD. This recombination energy stems from Source electrons that reoccupy empty states left behind by electrons previously tunnelling through the Emitter QD into the edge. The insight that such Auger-like recombination processes cause the unexpected currents is underpinned by additional measurements with a Sensor QD that solely detects currents generated by such processes, and through a theoretical analysis employing non-equilibrium perturbation theory. Our findings indicate that inter-edge interactions play a significant role in quantum Hall edge channel equilibration, and constitute an energy loss mechanism that has so far been disregarded.

## Results

**QD electron spectrometer.** A scanning electron micrograph of a typical sample to investigate the relaxation in quantum Hall edge channels is shown in Fig. 1a. Metallic top-gates are used to electrostatically define QDs in a high-mobility two-dimensional electron system (2DES) incorporated 90 nm below the surface of a GaAs/AlGaAs heterostructure. The electron mobility is $\mu_{el} = 2.2 \times 10^6$ cm$^2$ V$^{-1}$ s$^{-1}$ at $T = 1.3$ K with an electron density of $n_s = 2.0 \times 10^{11}$ cm$^{-2}$. Three QDs labelled Emitter (red), Detector (blue) and Sensor (green) are defined. When a magnetic field is applied perpendicular to the plane of the 2DES, chiral electron transport along the sample edge will be present, as indicated with yellow arrows. Ohmic contacts labelled Source, Drain, Reservoir, *Right* and *Left* allow us to apply DC voltages as indicated. The currents flowing into these terminals of the device are measured using current-voltage converters. Current flowing into a terminal

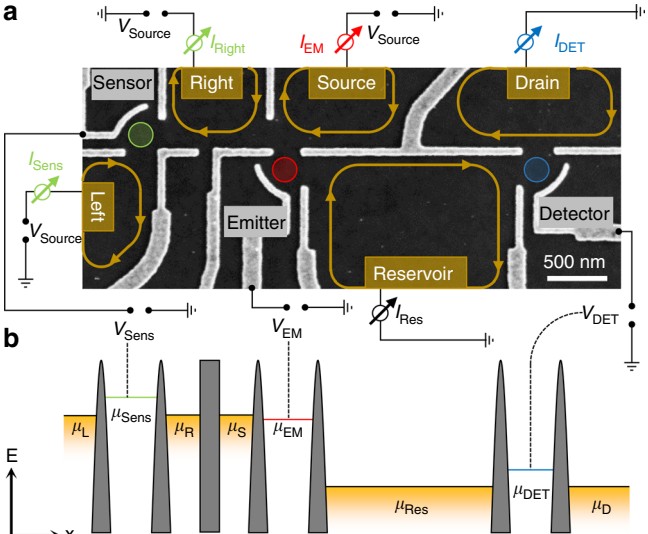

**Fig. 1** Quantum dot spectrometer in the quantum Hall regime. **a** Scanning electron micrograph of a typical sample to investigate energy relaxation in the quantum Hall edge channels. Metallic top-gates (light-grey) on the surface of the GaAs (dark grey) are used to define QDs (indicated by circles). A magnetic field is applied perpendicular to the 2DES (except in Fig. 2). The chirality of the resulting edge-channels is indicated by arrows. The current through each QD is measured separately in the contacts (yellow patches). **b** Energy schematic of the sample depicted in (**a**)

can lift the voltage applied to the 2DES by a few microvolts, due to the combined resistance of the 2DES, contacts and measurement lines (total resistance is typically 20 kΩ). All measurements were performed in a dilution refrigerator at the electronic base temperature $T_{el} = 25$ mK. In a typical experiment, current is injected from the Source through the Emitter QD into the Reservoir. The Detector and Sensor QDs are used to measure the edge-excitations of the Reservoir and *Right* contact, respectively, caused by the injected electrons. The barrier gate between the Source and *Right* regions of the 2DES is tuned such as to effectively suppress electron transfer between the two regions.

Figure 1b schematically illustrates the alignment of the different electrochemical potentials ($\mu_i$) under the conditions of our experiment. The electrochemical potentials of the Fermi seas (yellow in Fig. 1b) are kept constant while the electrochemical potentials of the QDs are varied by changing the voltages applied to the respective plunger gate. Finite bias spectroscopy of the individual QDs allows us to relate the plunger gate voltage to the energy of the QD electrochemical potential quantitatively. In such a configuration, resonant single-electron transfer from the Source to the Reservoir through the Emitter QD creates a single hole and a single-electron excitation in the edge channels of the respective regions. The non-equilibrium single-electron excitation, for example, propagates along the sample edge. If it remains unaffected by interactions with other electrons it can be detected by the Detector QD at the injection energy. However, if it interacts on its way with other electrons, it loses energy and causes an edge channel shake-up.

The result of such a transfer experiment from the Emitter to the Detector QD at zero magnetic field, i.e., in the absence of chiral edge transport, is shown in Fig. 2. A constant voltage $V_{Source} = -400$ μV is applied between the Source and Reservoir contact. The currents through the Emitter QD (in red) and through the Detector QD (in blue) for varying Detector QD plunger gate voltage, i.e., for varying $\mu_{DET}$, are shown in Fig. 2.

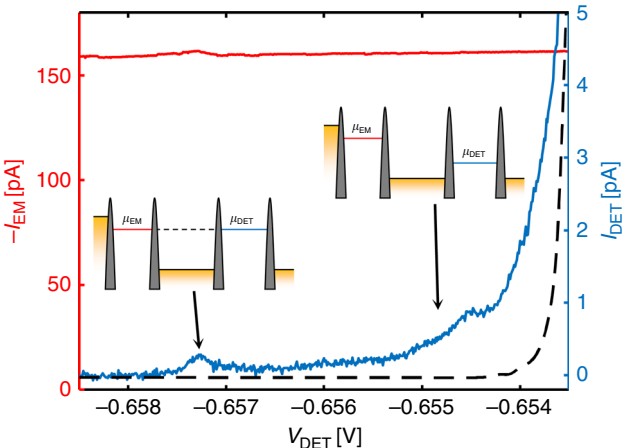

**Fig. 2** Zero-field electron transfer. Current through the Emitter (red) and Detector (blue) QD as a function of Detector plunger gate voltage in the absence of a perpendicular magnetic field ($B = 0$ T). The black dashed line indicates the background contribution of the Detector QD current due to an experimentally unavoidable slight misalignment between $\mu_{Res}$ and $\mu_{Drain}$. The current through the Emitter QD stays constant for varying Detector QD chemical potential. The current through the Emitter QD shows both elastic and inelastic contributions. The insets show the schematic level alignment of the respective configurations

The black dashed line shows the current through the Detector QD in the absence of a current through the Emitter QD. It corresponds to the background contribution of the Detector QD current due to an experimentally unavoidable slight misalignment between $\mu_{Res}$ and $\mu_{Drain}$. Here, the Emitter electrochemical potential is kept constant inside the bias window, i.e., the emission energy of the electrons ($\mu_{EM}$) is fixed, only $\mu_{DET}$ is varied. The current through the Emitter QD stays roughly constant as a function of Detector QD plunger gate. This is expected due to the small cross-capacitive coupling between Detector and Emitter QD.

At large negative values of $V_{DET}$ (around $V_{DET} = -0.6573$ V) the resonance condition $\mu_{EM} \approx \mu_{DET}$ holds and we measure a peak of elastic electron transfer between the QDs (see insets for the corresponding level alignment). The elastic transfer probability ($|I_{DET}/I_{EM}|$ at $\mu_{EM} = \mu_{DET}$) is rather small, here on the order of 0.2%. It depends on the emission energy and the tunnelling rates. At more positive values of $V_{DET}$ the detection energy is lower than the emission energy ($\mu_{DET} < \mu_{EM}$) and the shake-up of the Fermi sea is measured. The Detector current increases for lower detection energies close to $\mu_{DET} = 0$. The ballistic transfer of electrons between QDs, theoretically analysed in a different context in ref. [17,18], has already been measured previously and shows quantitatively similar results[19,20].

**Spectroscopy of edge channel transfer.** In the absence of a magnetic field electrons can scatter back from the Detector QD and form a standing wave in the Reservoir 2DES area. Standing waves have been shown to alter transport properties drastically[21–23]. Applying a strong perpendicular magnetic field does not only reduce standing waves by reducing backscattering but also introduces chiral transport along the sample edge and favours the directed transfer of electrons. For all the following measurements a perpendicular magnetic field $B = 3$ T is applied, corresponding to the quantum Hall plateau at filling factor $\nu = 3$. In the regime, where the magnetic field is strong enough to support quantum Hall edge currents (here $B \geq 1$ T), we do not find a qualitative influence of the magnetic field strength or the

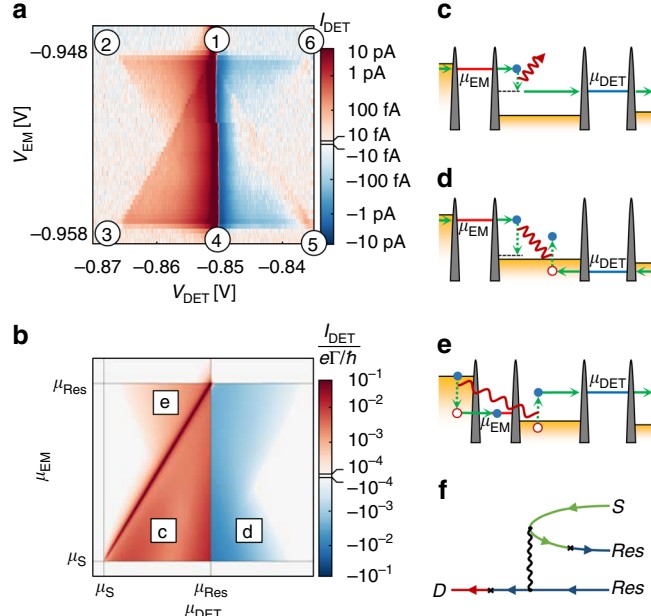

**Fig. 3** Spectroscopy of quantum Hall edge channel transfer. **a** 2D-colour plot of the current through the Detector QD for varying Detector QD ($x$-axis) and Emitter QD plunger gate voltage ($y$-axis). The encircled numbers refer to the energetic points mentioned in the text and are explained in detail in the Supplementary Note 1. The signal is plotted on a logarithmic colour scale preserving the direction of electron tunnelling in the colour code. **b** Calculated current through the Detector QD for the experimental situation depicted in (**a**). $\Gamma$ is the tunnelling rate through the QD. **c–e** schematic description of the electron scattering processes involved in the transfer. **f** Diagram corresponding to the process in (**e**), generating current in the triangle ①–②–④

bulk filling factor on the results presented below (see Supplementary Note 2). The QDs are coupled mostly to the outermost edge channel, corresponding to the energetically lowest Landau level. The presence of the outermost edge channel is not influenced by the bulk filling factor or the compressibility of the bulk. As for each different value of magnetic field the voltages applied to the tunnelling barrier gates have to be adjusted slightly, a quantitative analysis of the influence of the magnetic field strength on our measurement results is not possible.

Changing the values of both $V_{DET}$ and $V_{EM}$ during a transfer measurement, we measure the 2D-colour map of Detector current shown in Fig. 3a. Here, $V_{Source} = -400$ μV is applied between the Source and Reservoir contact. The signal is plotted on a logarithmic colour scale preserving the direction of electron tunnelling. A red signal corresponds to electrons tunnelling from the Reservoir lead to the Drain side (from Drain to Reservoir for a blue signal). Each point in the 2D-map corresponds to specific energies of the Emitter and Detector QD levels. Characteristic points are indicated by numbers and the corresponding energy level alignments are shown in Supplementary Note 1. The current through the Emitter QD for the same measurement (see Supplementary Fig. 1a) depends only on $V_{EM}$ and is independent of $V_{DET}$.

The peak of elastic transfer shown for the zero magnetic field case in Fig. 2a is largely suppressed in the Detector current in Fig. 3a at finite magnetic field. The diagonal line connecting ① and ③ still marks the boundary between a region of suppressed and sizeable Detector current. At points ①/③ we emit and detect electrons close to the Fermi energy of the Reservoir/Source contact.

Emitted electrons can interact with other electrons (see Fig. 3c, d) and thereby dissipate energy. Hence, a non-equilibrium distribution function will develop along the course of the outermost edge channel. This non-equilibrium current can be observed as long as the detection energy is lower than the emission energy and the Detector QD level is above the electrochemical potentials of the grounded contacts ($\mu_{EM} > \mu_{DET} > \mu_{Res/D}$; corresponding to the region inside the triangle ①–③–④).

The Detector QD level is equal to the equilibrium electrochemical potential of its connecting leads along line ① to ④. Lowering the Detector QD level further (more positive $V_{DET}$) will fill the Detector QD with an electron, hence enabling us to probe the Reservoir non-equilibrium distribution below its equilibrium electrochemical potential. The shake-up of the Fermi system will generate unoccupied states there exceeding the number of thermally unoccupied states present in equilibrium.

The maximal energy an emitted electron can lose in a scattering event is its energy above the Fermi energy, which we call the emission energy. This is also the maximal amount of energy which can be transferred to another electron, hence, the lowest lying state which will be emptied lies exactly this amount of energy below the Fermi level. The symmetric outline of the negative signal we observe supports this claim. Within the triangle ①–④–⑤ we hence probe "holes" in the Reservoir, generated by collisions of emitted electrons with Fermi sea electrons in the Reservoir (Fig. 3d). The slight positive signal close to the diagonal line ①–⑤ is due to the presence of an excited state in the Detector QD, which we will not discuss further here (see Supplementary Note 3 for details).

Around points ② and ⑥ we also observe a large shake-up of the edge channel distribution, however, the previously described processes cannot account for the involved energies. Around ② the detection energy is much higher than the emission energy. We attribute these high energy electrons to be the result of an Auger-like recombination process occurring in the course of electron tunnelling through the Emitter. An electron tunnelling through the Emitter will leave behind an empty state in the Source Fermi sea. This "hole" will dissipate its excess energy, possibly by recombining with an energetically higher lying electron in the same contact region. This recombination can, like in Auger recombination, excite another electron in the Reservoir region (Fig. 3e). The largest energy which could be transferred in such a process corresponds to the energy difference of the Source Fermi level to the Emitter QD level (explaining the negative slope diagonal from ② to ④). The excited electron will, due to the edge channels, be transported to the Detector QD where it is spectroscopically investigated. The negative signal around ⑥ can be understood as a consequence of "holes" in the Reservoir generated in the same process.

To substantiate that electron collisions cause the measured inelastic currents displayed in Fig. 3a, the currents in the triangles described above are set in relation to underlying physical processes by second order non-equilibrium diagrammatic perturbation theory. The Source, Drain and Reservoir regions are modelled as single, parallel and one-dimensional channels with linear dispersion. Tunnelling connects these channels via Emitter and Detector QDs represented by single resonant levels.

Dressing electron lines with tunnelling events in the self-energies of the Reservoir and the Source–Reservoir interaction generates distinct diagrams for processes which contribute to the inelastic current. The Detector current obtained by this approach is displayed in Fig. 3b. A finite-range model interaction between electrons in the Reservoir generates current in triangles ①–③–④ and ①–④–⑤. The same form of interaction between Source and Reservoir electrons, additionally accounting for the spatial separation of the respective channels, generates current in triangles ①–②–④ and ①–④–⑥. Figure 3f shows an exemplary diagram which corresponds to the transition amplitude of the Auger-like process depicted in Fig. 3e, and contributes in triangle ①–②–④.

While second order perturbation theory does not support strong enough interactions to account for features such as the largely absent line of elastic transfer in Fig. 3a (the trend of diminishing elastic current is captured, however, within the energy range defined by the Source–Drain bias, second order perturbation theory breaks down for higher values of interaction strength or transfer time), the treatment does show that Auger-like recombination in the Source causes signals in regions of the Emitter–Detector energy diagram, which cannot be explained in terms of interactions between Reservoir electrons alone.

**Direct measurement of Auger-like processes.** We now turn our attention to additional experimental investigations consolidating the interpretation of the data suggested above. The sample shown in Fig. 1a is explicitly designed to investigate the non-equilibrium distribution of electrons while emitting electrons through a QD. In addition to the two aforementioned QDs there is a third, the Sensor QD. The Sensor QD is electrically isolated from the Emitter/Detector part of the sample, which means that no electric current can flow from Source to Right (see Supplementary Note 4). The gates between the Source and the Right contact are biased with a large negative voltage which induces a barrier in the 2DES. The Source bias voltage (here $V_{Source} = -700\,\mu V$) is also applied to both the Right and the Left lead of the Sensor QD (see Fig. 1b), i.e., $\mu_S = \mu_R = \mu_L$. In the following we use the Sensor QD to measure the non-equilibrium distribution in the Right contact while electrons tunnel through the Emitter QD. Such a measurement is shown in Fig. 4. The current through the Sensor QD, now plotted as a function of $V_{EM}$ and $V_{Sens}$, is shown in Fig. 4a. For a plot of the current through the Emitter, as well as for the level schematics corresponding to the highlighted points, we refer to the Supplementary Fig. 3. The slight tilt in the data, seen for example along line ②–⑥, is due to the cross-capacitance between the Sensor and Emitter QDs and their respective plunger gates, which is larger than for the Emitter and Detector. In the current through the Sensor QD (in Fig. 4a) we observe a positive current in the triangle ①–②–④ and a negative current in the triangle ①–④–⑥. The signal in the triangle ①–③–④ is mostly absent, except for a slight positive current along the line connecting ③ and ④. Similarly, also the signal in the triangle ①–④–⑤ is absent, except for a slight negative current along the line connecting ④ and ⑤. Away from the line connecting ① and ④ the current through the Sensor QD is directly proportional to the non-equilibrium distribution of the outermost edge channel in the Right contact.

From the signal measured in the Sensor we draw two major conclusions. First, measuring a signal away from the background current along the line connecting ① and ④ shows that there are non-equilibrium electrons generated in the Right contact region while electrons tunnel through the Emitter QD. As the direct electron transport from the Source to the Right side is barred, we observe an energy transfer from the Source contact to the Right contact. Second, the fact that the maximal energy at which electrons are observed to tunnel through the Sensor depends on the energetic position of the Emitter level ($\mu_{Sens}^{max} - \mu_S = \mu_S - \mu_{EM}$; indicated by the dashed line connecting ② and ④ in Fig. 4a) excludes a thermopower effect to be the reason for this observation, as heating would be independent of the Emitter chemical potential. However, Auger-like recombination can account for this observation.

The signal close to ③ and ⑤, does not comply with the Auger-like recombination in the Source Fermi sea, but can be explained

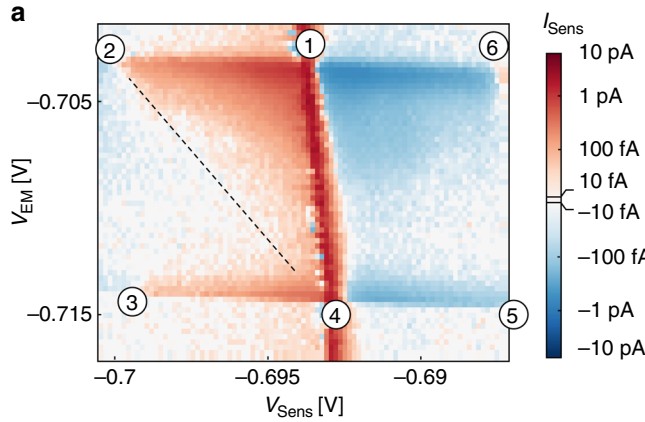

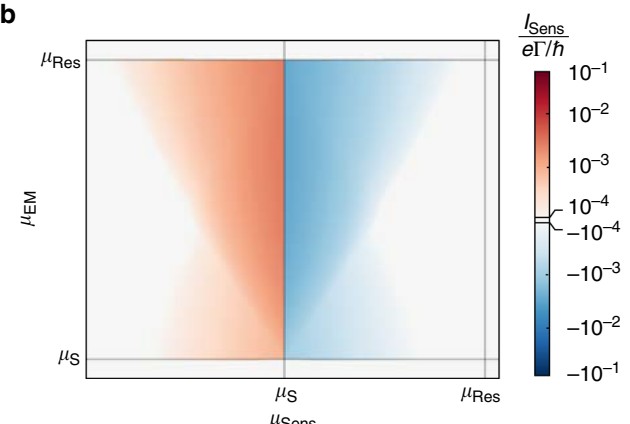

**Fig. 4** Direct spectroscopy of Auger-like processes. **a** 2D-colour plot of the current through the Sensor QD for varying Sensor QD (*x*-axis) and Emitter QD plunger gate voltage (*y*-axis). The signal is plotted on a logarithmic scale preserving the direction of electron tunnelling in the colour code. The dashed line corresponds to the maximal energy which can be transferred by Auger-like recombination: $\mu_{Sens}^{max} - \mu_S = \mu_S - \mu_{EM}$. **b** Calculated current through the Sensor QD for the experimental situation depicted in (**a**). $\Gamma$ is the tunnelling rate through the QD

capacitively coupled to the emitted electrons. While global energy conservation is fulfilled, our measurements indicate that highly biased quantum devices can excite degrees of freedom in other nearby systems which are not directly but only capacitively coupled. To attain a more complete picture of energy relaxation, future experimental and theoretical analyses accounting for multiple edge channels, allowing to control such multi-channel Auger processes, should be carried out. Furthermore, investigating such energy redistribution in the fractional quantum Hall effect might shed light on the nature and interactions of quasiparticles in fractional quantum Hall ground states.

In the course of the preparation of this paper we became aware of a similar study on electron transfer between QDs[24].

### Data availability
The data that support the findings of this study are available from the authors on reasonable request.

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

by an Auger-like recombination in the Reservoir Fermi sea. We have seen that an emitted electron undergoes scattering events where another electron is excited. This could account for the signal measured at positions ③ and ⑤.

To underpin the aforementioned conjectures derived from the measurement data displayed in Fig. 4a in the extended experiment, the theoretical model is amended by a *Left* and a *Right* channel as well as by a Sensor resonant level. The Sensor current obtained from the amended model is shown in Fig. 4b. Interactions between Source and *Right* electrons here indeed generate current in triangles ①–②–④ and ①–④–⑥, while interactions between Reservoir and *Right* electrons generate current in triangles ①–③–④ and ①–④–⑤. The smaller magnitude of the current in the latter triangles results from the larger separation of the channels involved in the respective transport processes.

### Conclusions
In conclusion, we have investigated the non-equilibrium distribution of electrons generated by electron emission from a QD into a quantum Hall edge channel. We observe both the direct shake-up of the edge channel into which electrons are emitted, and the shake-up of distant edge channels which are only

24. Rodriguez, R. H. et al. *Strong Energy Relaxation of Propagating Quasiparticles in the Quantum Hall Regime*. Preprint at http://arxiv.org/abs/1903.05919 (2019).

## Acknowledgements

This work was supported by the Swiss National Science Foundation through the National Center for Competence in Research (NCCR) Quantum Science and Technology (QSIT). S.G.F. acknowledges financial support from the Minerva foundation. Y.G. was supported by DFG RO 2247/11-1 and CRC 183 (project C01), and the Italia–Israel project QUANTRA. Y.M. acknowledges support from ISF grant 292/15.

## Author contributions

T.K. fabricated devices, conceived and performed the experiment and analysed the data with supervision from T.I. and K.E. M.R. supported the experiments and helped understand the data. C.R. and W.W. grew the heterostructure. Y.G. and Y.M. initiated the theoretical calculation, and devised with S.G.F. the model. S.G.F. carried out the calculation. The paper was written by T.K. and S.G.F. with input from all authors.

## Additional information

**Competing interests:** The authors declare no competing interests.

