## [Peer Review File · Nature Communications]

Reviewers' comments:

Reviewer #1 (Remarks to the Author):

The authors reported,

- i. Spectroscopy of zero-field electron transfer utilizing quantum dots as energy filters
- ii. Spectroscopy of edge channel transfer
- iii. Observation of Auger-like recombination process
- ix. Direct measurement of non-local Auger-like process utilizing well-separated 2DES
- x. The results are (partially) reproduced by the non-equilibrium perturbation theory

I think that these are novel and of interest to readers in the community and the wider field. However, the following issues should be answered before publication because some relate to the main points of this paper.

As to the measurement setup,

1. What are the values of the Ohmic contact resistance (at the zero magnetic field and in the quantum Hall regime) and the input impedance of the current measurement circuits?

The resistance modifies the actual voltage applied to the 2DES depending on the current.

For example in the case of Reservoir 2DES,

$$V_{\text{Res}} = (R_{\text{Res_Ohmic}} + R_{\text{Res_meas_circuit}}) * I_{\text{Res}}$$

I guess that the effect is small, but please give the information to readers.

As to the zero-field electron transfer,

2. In Fig.2 at the resonant condition ($V_{\text{DET}} = -0.6573\text{V}$), an increase of $-I_{\text{EM}}$ as a function of V_{DET} is observed.

The increase of $-I_{\text{EM}}$ ($\sim 3 \text{ pA}$) is larger than the increase of I_{DET} ($\sim 0.3 \text{ pA}$).

What is the mechanism of this increase?

As to the spectroscopy of edge channel transfer,

3. Page 3 Line 3 and 94: Why the elastic transfer peak is largely suppressed, and the interaction is so strong in this measurement?

Is the observed relaxation length ($<$ the distance between Emitter QD and Detector QD ~ 2 micrometers) consistent with the previous reports like Ref.[8]?

4. Fig.3a: Adding labels I, II and III at corresponding positions of (V_{DET} , V_{EM}) will help readers to understand.

As to the direct measurement of the Auger-like process

5. Showing the observed $I_{\text{Sens}} + I_{\text{Right}} \ll 10 \text{ fA}$ in the measurement range of (V_{Sens} , V_{EM}) supports that there is no charge transport between Source and Right.

6. Fig.4: Adding schematics like Fig.3c will help readers to understand.

7. Fig.4b: The label of the lower axis might be μ_{Sens} .

8. Extended Data Fig.1c and 3c: Adding schematics of the movement of electrons and holes like in Main text Fig.3c will help readers to understand.

As to the second-order non-equilibrium diagrammatic perturbation theory,

9. How do the authors obtain the coupling strength between the channels used in the perturbation theory?

By calculating the capacitances between the channels from the gate geometry?

10. What is the definition of Γ in Fig.3b and 4b?

The tunneling rate through Emitter QD?

Are the resulting I_{DET} and I_{Sens} consistent with the observed I_{DET} and I_{Sens} ?

Reviewer #2 (Remarks to the Author):

In this paper the authors perform and analyse an experiment on non-equilibrium edge spectroscopy in quantum Hall edge states. They argue that capacitive coupling between different parts of the system leads to Auger-like non-local processes which result in energy exchanges between distant parts of the system. The experimental results are compared with theoretical predictions obtained from non-equilibrium perturbation theory.

The question of "missing energy" in non-equilibrium spectroscopic measurements of quantum Hall edge states (as e.g. observed in experiments by F. Pierre et.al) is interesting and important. This phenomenon has been, at least partially, explained in ref [15] and in subsequent paper by the same authors. However (private communication with F. Pierre) this explanation may have not accounted for other mechanisms.

In present paper the authors suggest another possible mechanism, which may be responsible for "missing energy" phenomenon, and which should be important in analysing other non-equilibrium experiments with QHE edge states, and therefore I recommend the paper for publication in Nature Communications.

Comments:

1. It is quite surprising that elastic contribution is strongly suppressed in the data for the case in presence of magnetic field (as compared to zero-field case), and it would be good to have some suggestion on explaining why this is so.
2. The authors suggest that their results do not depend much on the filling factor (and have not discussed capacitive coupling between different edge modes on the same edge), while the energy exchange mechanism in previous experiments was due to capacitive coupling between co-propagating edge states. Could the authors perhaps comment on this.
3. If possible, it would be useful if the authors present a bit more details on their theoretical model, and calculations together with parameters which they use in Figs 3b,4b.
4. The explanation of the results of this experiments as due to Auger-type processes seems to be plausible in this case. However, the geometry of the experimental setups by F. Pierre is quite different, and it would be useful to see some applications of the author's reasoning in explaining the missing energy phenomena in these experiments.

Reviewer #3 (Remarks to the Author):

This work shows strong evidences for energy transfers involving quantum Hall channels located along different edges separated by metallic gates at the surface, which is usually neglected. This experimental investigation is particularly relevant in the electron quantum optics context, where preserving the quantum coherence of electrons propagating along the quantum Hall edges is essential but very challenging and impeded by mechanisms yet not clearly established. Using a quantum dot for both emission and energy spectroscopy allows for a very distinguishable signature of inter-edge energy transfers, appearing as inverted triangles in the V_{EM} V_{DET} color plots that are incompatible with energy redistributions along the same edge. I find that the paper is well-written, reporting on clear experimental findings, and that the interpretation is sound. I recommend publication in Nature Communications provided the points below are adequately

addressed.

1. Abstract, L18: "can be untangled" should be replaced by "could be untangled", to emphasize that this remains a possibility and not a certainty (for explaining the extra dissipation seen in other experimental setups). In particular, the energy was found to be conserved along the propagation path when closing the inner channel into loops at $\nu=2$ (Altimiras et al, PRL 105, 226804 (2010)), which implies that the distant coupling to other propagative edge channels was small in this experiment. Yet unexpected losses were observed with two copropagative channels along the same path.

2. Page 2, L52-55 and Figure 2: the authors show as a dashed line the current resulting from the misalignment between μ_{res} and μ_{det} . From the text it seems that this misalignment is known and injected into the quantum dot model to calculate the dashed line. If this is the case, could the authors provide the value of this misalignment and explain how it was obtained experimentally. If not, please explain more precisely what represents this dashed line.

3. Page 2, L92-95: The authors state they did not find a qualitative influence of the bulk filling factor. But does this statement include the bulk filling factor $\nu=2$, considered the most canonical and where most mentioned electron quantum optics experiments were performed? Can the authors explicitly state whether this Auger-like behavior was also seen at $\nu=2$? If this is the case, I believe it is important to show some $\nu=2$ data, at least in the supplementary. If not, please explicitly point this out as the text is presently ambiguous.

4. The value of the source voltage is provided for the data in fig 2 and fig 4 but not fig3. Could the authors add this possibly important information? Could the authors also explicitly indicate if they observe a dependence of the Auger signal/direct signal ratio as a function of V_{source} ? Intriguingly, in extended data fig 4, this ratio seems to increase with V_{source} to the point of being larger than 1 or at least comparable to 1, although there is also the additional complication of having more excited states within the energy window.

5. Page 3, L41-45: It is mentioned that the positive signal along the 1-5 line of fig 3a results from the presence of an excited level in the detector quantum dot (as discussed in the supplementary). What is the energy of this level? Is there only one additional level? It would be extremely useful to see a level spectroscopy of this quantum dot for the same configuration used for the data of fig. 3a.

6. The second section of the supplementary information discusses the effect of additional quantum dot levels. In addition to this discussion, it would be most straightforward and convincing to directly show the result of the calculated current using the standard master equation for sequential tunneling with the known position of the levels (adjusting the different tunneling rates) and the calculated energy distribution shown in fig 3b.

We thank all Reviewers for their comments and suggestions to improve the present manuscript entitled: "Auger-spectroscopy in quantum Hall edge channels: a possible resolution to the missing energy problem".

Below we provide answers to all of the reviewers' questions. These answers as well as the corresponding corrections are highlighted in the manuscript in red.

Reviewer #1:

1. What are the values of the Ohmic contact resistance (at the zero magnetic field and in the quantum Hall regime) and the input impedance of the current measurement circuits?

The resistance modifies the actual voltage applied to the 2DES depending on the current.

For example in the case of Reservoir 2DES,

$$V_{\text{Res}} = (R_{\text{Res_Ohmic}} + R_{\text{Res_meas_circuit}}) * I_{\text{Res}}$$

I guess that the effect is small, but please give the information to readers.

Answer: We did not systematically investigate the Ohmic resistance in the device presented in the manuscript, however, typical values are 2-3 kOhm per contact. In combination with the edge channel resistance at $\nu=3$ we end up with a typical resistance of 10kOhm. Our measurement lines have additional 10kOhm resistors for filtering purposes, leading to a total of 20kOhm. This value has to be compared to the resistance of the QD, which is typically on the order of 1MOhm (200 μ V bias Voltage for 200 pA current). This means the effect is on the few percent level. In the present scenario the Reservoir 2DEG would be lifted by 4 μ V. This effect is slightly visible in the measurements, as the Detector plunger gate voltage of the Detector QD resonance shifts slightly once current is flowing through the Emitter QD. (The effect is seen, e.g. in Extended Data Figures 1, 4, and 5, where the resonance position shifts slightly when current flows through the Emitter QD.)

The extracted value of 4 μ V is on the same order of magnitude as the drift we experience on the bias voltages applied to the 2DES. (See more on that in the answer to Reviewer#3 Question 2)

A sentence has been added in the manuscript, page 2 lines 14-18 to highlight the finite contact resistance.

2. In Fig.2 at the resonant condition ($V_{\text{DET}} = -0.6573\text{V}$), an increase of $-I_{\text{EM}}$ as a function of V_{DET} is observed.

The increase of $-I_{\text{EM}}$ (~ 3 pA) is larger than the increase of I_{DET} (~ 0.3 pA).

What is the mechanism of this increase?

Answer: In the absence of a magnetic field we observe that the density of states in the Reservoir area is modulated due to the confining top-gate structure, such that standing waves can form. We observe a high elastic transfer probability when both the Emitter and Detector QD couple to the same standing wave mode (i.e. we see a well pronounced elastic peak, corresponding to the situation shown in Fig. 2). The electrons that are transferred through the Detector QD cause a non-zero average occupation of the Detector which due to the capacitance between the Detector QD and the Reservoir will slightly shift the energy of these standing waves in the Reservoir. In the present situation the standing waves are shifted in such a way that the Emitter QD has slightly increased coupling to the Reservoir which cause an increased current (also the opposite effect has been observed)

This finding has led to a more detailed investigation of these standing waves in 2DEGs, i.e.:

C. Rössler, et al., PRL 115, 166603 (2015) (Reference 22 of the manuscript)
R. Steinacher, et al., PRB 98, 075426 (2018) (Reference 23 of the manuscript)

3. Page 3 Line 3 and 94: Why the elastic transfer peak is largely suppressed, and the interaction is so strong in this measurement?

Is the observed relaxation length ($<$ the distance between Emitter QD and Detector QD ~ 2 micrometers) consistent with the previous reports like Ref.[8]?

Answer: In Reference [8] significant relaxation is detected starting from propagation lengths of 4 micrometer and onwards. However, it is hard to draw conclusions comparing these results to ours, as in this case emission of electrons was achieved by a quantum point contact and detection by a quantum dot while in our experiment we report on quantum dot to quantum dot transfer. The available phase space for electron scattering is much larger when electrons are emitted by a quantum dot, i.e. all states between the emission energy and the Fermi energy of the Reservoir are empty, while in the quantum point contact emission they are (at least partially) occupied.

Our observations are however consistent with a new quantum dot transfer experiment by the same group [ArXiv:1903.05919], where at a dot separation of about 2.17 micrometers no elastic peak is observed at filling factor $\nu = 2$.

At our filling factor $\nu = 3$, relaxation is supposedly even stronger due to the presence of the additional decay channel provided by the third edge channel.

We added a sentence at the end of the manuscript, highlighting the recent results of ArXiv 1903.05919.

4. Fig.3a: Adding labels I, II and III at corresponding positions of (V_{DET} , V_{EM}) will help readers to understand.

This question will be answered in combination with questions 6 and 8

6. Fig.4: Adding schematics like Fig.3c will help readers to understand.

8. Extended Data Fig.1c and 3c: Adding schematics of the movement of electrons and holes like in Main text Fig.3c will help readers to understand.

Answer: We aimed to achieve a clear distinction between characteristic points in the plunger-plunger map corresponding to special alignments of the quantum dot energy levels (denoted by encircled Arabic numbers) and transfer processes happening not just at a single point in this map but in entire regions of the plunger-plunger space (denoted by roman numbers). For this reason we would like to keep the existing schematics. We however added Roman labels I-III in Fig. 3.b, as well as process schematics in Extended Data Figure 3 as called for in question 6 and 8.

5. Showing the observed $I_{Sens} + I_{Right} \ll 10$ fA in the measurement range of (V_{Sens} , V_{EM}) supports that there is no charge transport between Source and Right.

Answer: A new section "Electrical isolation of the sensor QD" together with Extended Data Figure 6 have been added to the supplemental material.

A reference to the Supplemental has been added at the corresponding spot in the main text

(page 4 line 10)

The measured sum of $I_{\text{Sens}}+I_{\text{Right}}$ is slightly larger than 10fA, but characteristically $<100\text{fA}$. This is due to the measurement setup and our Current-voltage (IV) converters. The IV Converters are built in the breakout box of our cryostat and are all in the same shielded box. This ensures small input capacitances and capacitive noise gain, but comes at the cost that the outputs of the IV-converters are unshielded with respect to each other. The slightly elevated signal we see in the $I_{\text{Sens}}+I_{\text{Right}}$ at $V_{\text{EM}} -0.704$ to -0.710V corresponds to the region of high current through the Emitter quantum dot (Extended Data Figure 3). This means there is a capacitive coupling of the IV converter output of the Emitter QD to the IV converter output of either I_{Sens} or I_{Right} . The corresponding numbers are as follows: the 700pA measured for the emitter correspond to 7V on the output (10GOhm feedback resistor), while the 100fA error signal corresponds to an error of 1mV, the noise limit of 10fA would correspond to 100 μV on the output of the IV converter.

6. Fig.4: Adding schematics like Fig.3c will help readers to understand.

Answer: Please see our answer to Reviewer #1 Question 4 addressing this issue.

7. Fig.4b: The label of the lower axis might be μ_{Sens} .

Answer: Yes indeed, the label has been adjusted.

8. Extended Data Fig.1c and 3c: Adding schematics of the movement of electrons and holes like in Main text Fig.3c will help readers to understand.

Answer: Please see our answer to Reviewer #1 Question 4 addressing this issue.

9. How do the authors obtain the coupling strength between the channels used in the perturbation theory?

By calculating the capacitances between the channels from the gate geometry?

Answer: The absence of significant current on the line which describes elastic transfer at equal Detector and Emitter energies in the strong magnetic field indicates that the actual interaction strength in the experiment exceeds the interaction strength admissible in the perturbative approach. This is pointed out in the last paragraph of the third Section (spectroscopy of edge channel transfer), and additional details on the employed model interaction are now provided in a new section added to the supplemental material. The precise limitations of perturbation theory will furthermore be discussed in detail in a standalone theory paper. The value for the interaction strength used to generate Figs. 3b and 4b is a value that is still supported by the perturbative calculation.

10. (part1) What is the definition of Γ in Fig.3b and 4b?

The tunneling rate through Emitter QD?

Answer part1: The definition of Γ , which indeed corresponds to the tunneling rate through the dots, as well as of all other parameters that appear in the theoretical model, alongside the employed parameter values, are now listed in Table 1 that has been added in a new section of the supplemental material.

(part2) Are the resulting I_{DET} and I_{Sens} consistent with the observed I_{DET} and I_{Sens} ?

Answer part2: To achieve quantitative agreement with the experimental data, it would be necessary to take into account many more factors that escape the scope of the current theoretical treatment (and, in part, all other theoretical approaches known to us), such as higher interaction orders, sample geometry, edge channels on the same edge, non-zero temperature, as well as interactions between the quantum dots and the channels of the sample. Our theoretical model serves to capture the essential features of the Auger-like inter-channel recombination effects, to build up upon in the future by including aforementioned factors. At the present stage and complexity, it is however already possible to achieve a comparable ratio between inelastic I_{DET} to injected current I_{EM} as is observed in the experiment.

Reviewer #2:

1. It is quite surprising that elastic contribution is strongly suppressed in the data for the case in presence of magnetic field (as compared to zero-field case), and it would be good to have some suggestion on explaining why this is so.

Answer: We can compare the zero field case to the case of finite magnetic field in a phenomenological manner. From the answer of Reviewer#1 Question 2 we know that the high elastic transfer probability is at energies where a standing wave couples well to both quantum dots. In a magnetic field the time-reversal symmetry is broken and the standing waves are absent. One can also expect that the time of flight of the electron from Emitter to Detector increases in the case of quantum Hall edge transport (Fermi velocity vs. $E \times B$ -drift velocity), thus increasing the time of interaction of the transferred electrons. All of these effects can lead to a decrease in elastic transfer.

Our results are however consistent with a new quantum dot transfer experiment [ArXiv:1903.05919]. See also Answer to Reviewer #1 Question 3 where we elaborate further on that.

2. The authors suggest that their results do not depend much on the filling factor (and have not discussed capacitive coupling between different edge modes on the same edge), while the energy exchange mechanism in previous experiments was due to capacitive coupling between co-propagating edge states. Could the authors perhaps comment on this.

Answer:

Our focus of the theory is indeed not placed on interactions between different edge modes on the same edge, which has been investigated in detail experimentally e.g. in Refs. [8,16] and theoretically e.g. in Refs. [13-15]. The reason is that these interactions cannot account for the signals in triangles (1)-(2)-(4) and (1)-(4)-(6) that are due to Auger-like inter-edge recombination which is treated in our theoretical description. In this theoretical model, the signals in triangles (1)-(3)-(4) and (1)-(4)-(5) are only qualitatively accounted for by interactions between electrons on the same sample edge, whereas the dominant relaxation mechanism is in this case expected to be inter-channel interaction on the same edge, as

pointed out by the referee. From the theoretical viewpoint, combining both, finite range inter-edge interactions and inter-channel interactions on the same edge, is a highly non-trivial task that requires further investigation which far exceeds the scope of the present work.

To show that the experimental data are qualitatively independent of the filling factor we added a new section to the supplemental showing data for filling factors $\nu=2$ and $\nu=4$, as asked for by Reviewer#3 Question 3.

3. If possible, it would be useful if the authors present a bit more details on their theoretical model, and calculations together with parameters which they use in Figs 3b,4b.

Answer: Following the referee's recommendation additional information on the interaction matrix elements used in the theoretical part are presented in a newly added section in the supplemental material, alongside the parameters used to generate the plots, which are presented in the newly added Table 1 in the Extended Data. The detailed presentation of the theoretical model and the calculation of the current shown in Fig. 3b and 4b will be presented in a standalone paper.

4. The explanation of the results of this experiments as due to Auger-type processes seems to be plausible in this case. However, the geometry of the experimental setups by F. Pierre is quite different, and it would be useful to see some applications of the author's reasoning in explaining the missing energy phenomena in these experiments.

Answer: In the setup by F. Pierre in Ref. [8] the current carrying edge channel passes several pinched off QPCs that are used to regulate the sample's propagation path (see Fig. 1 in the supplemental material *ibid.*). It is conceivable that, due to the relative proximity of these edge channels passed by the current carrying edge, energy is here lost to these parts of the sample. It would be interesting to probe such adjacent channels in a further experiment to test for the presence of excitations.

In this context, with regard to another experiment by F. Pierre [Tuning Energy Relaxation along Quantum Hall Channels; Altimiras, et al., PRL 105, 226804 (2010)], it would be particularly interesting to see whether relaxation is different when the right (yellow) QPC in Figs. 2 and 3 is pinched off. In the situation of Fig. 2, Auger-like processes should be able to excite electrons in the pinched off channel, causing relaxation in the initially populated channel. In the situation of Fig. 3, however, the pinched off channels, due to their short length, should be inaccessible also to Auger-like recombination, effectively inhibiting this relaxation mechanism for the originally populated channel.

Reviewer #3:

1. Abstract, L18: "can be untangled" should be replaced by "could be untangled", to emphasize that this remains a possibility and not a certainty (for explaining the extra dissipation seen in other experimental setups). In particular, the energy was found to be conserved along the propagation path when closing the inner channel into loops at $\nu=2$ (Altimiras et al, PRL 105, 226804 (2010)), which implies that the distant coupling to other propagative edge channels was small in this experiment. Yet unexpected losses were observed with two copropagative channels along the same path.

Answer: The abstract has been changed according to the comment of the referee.
Concerning the experiment of Altimiras, et al. we refer to the answer of Reviewer#2
Question 4.

2. Page 2, L52-55 and Figure 2: the authors show as a dashed line the current resulting from the misalignment between μ_{res} and μ_{det} . From the text it seems that this misalignment is known and injected into the quantum dot model to calculate the dashed line. If this is the case, could the authors provide the value of this misalignment and explain how it was obtained experimentally. If not, please explain more precisely what represents this dashed line.

Answer: The chemical potentials of the different regions of the sample are set by bias voltages applied over the input of an IV-converter. As such this voltage is not perfectly stable but has noise and drifts. A significant influence is the temperature of the electronics (room-temperature drifts) a 1 degree difference in ambient temperature can cause a voltage drift of a few microvolts on the input, which will be measurable in our setup. We regularly adjust these offsets, however not for every single measurement. Thereby, a difference in μ_{Res} and μ_{Det} can develop over time.

The dashed line in Fig.2 corresponds to the current measured through the Detector QD when no current is flowing through the Emitter QD. (in the case of Fig.3a, this corresponds to a linecut at the very top or very bottom of the Figure. The corresponding measurement without magnetic field [not shown] has to be considered for Fig.2)

The dashed line extracted in this way is a lower bound for the background contribution and does not take into account the shift of μ_{Res} due to the contact resistance to ground, explained in more detail in the Answer to Reviewer#1 Question1.

A sentence has been added to the manuscript, page 2 line 56-61

3. Page 2, L92-95: The authors state they did not find a qualitative influence of the bulk filling factor. But does this statement include the bulk filling factor $\nu=2$, considered the most canonical and where most mentioned electron quantum optics experiments were performed? Can the authors explicitly state whether this Auger-like behavior was also seen at $\nu=2$? If this is the case, I believe it is important to show some $\nu=2$ data, at least in the supplementary. If not, please explicitly points this out as the text is presently ambiguous.

Answer: We indeed observe the Auger-like recombination for varying filling factors, additional data for $\nu=2$ and $\nu=4$ is shown in the Extended Data in a new section.

A reference to the supplemental has been added to the manuscript, page 2 line 101

4. The value of the source voltage is provided for the data in fig 2 and fig 4 but not fig3. Could the authors add this possibly important information? Could the authors also explicitly indicate if they observe a dependence of the Auger signal/direct signal ratio as a function of V_{source} ? Intriguingly, in extended data fig 4, this ratio seems to increase with V_{source} to the point of being larger than 1 or at least comparable to 1, although there is also the additional complication of having more excited states within the energy window.

Answer: The corresponding Source voltage (400 μV) for Fig 3 has been added to the

manuscript (page 2 lines 113-115)

We have not observed a systematic ratio/dependence of the Auger vs. direct signal as a function of Source voltage. The effect visible in extended data Figure 4, mentioned by Reviewer#3 has two possible explanations. The first is simply due to the logarithmic color scale which highlights effects at low currents and obscures features at higher current values. This is of course chosen to highlight the faint features of the Auger-recombination at lower Source voltage. The second effect, which is very prominent in extended Data Figure 4 is the concomitant increase of the current through the Emitter (not shown, but similar to extended Data Figure 3a). One should thus not only compare the Auger signal to the direct signal, but also normalize it by the amount of current measured in the Emitter QD.

Taking all this into account we did not observe a systematic dependence of either direct to Auger signal nor of the positive to negative contribution of the transferred signal.

5. Page 3, L41-45: It is mentioned that the positive signal along the 1-5 line of fig 3a results from the presence of an excited level in the detector quantum dot (as discussed in the supplementary). What is the energy of this level? Is there only one additional level? It would be extremely useful to see a level spectroscopy of this quantum dot for the same configuration used for the data of fig. 3a.

Answer: There is only one (dominant) excited state in the measurement shown in Fig. 3a. The level spectroscopy of this dot is less conclusive than the measurements shown in the Extended Data. The excited state energy is approximately $500\mu\text{eV}$, its tunnel coupling to the leads is however orders of magnitude stronger than the ground state's. Therefore we can see a small effect even at a Source voltage of $400\mu\text{eV}$

6. The second section of the supplementary information discusses the effect of additional quantum dot levels. In addition to this discussion, it would be most straightforward and convincing to directly show the result of the calculated current using the standard master equation for sequential tunneling with the known position of the levels (adjusting the different tunneling rates) and the calculated energy distribution shown fig 3b.

Answer: We thank the reviewer for proposing the theoretical analysis of the role of excited detector states. Intuitively, the influence of an excited state in the detector can be understood in the following way: Both the lower lying and the excited detector state generate a transfer diagram similar to Extended Data Figure 2. The transfer diagram of the excited state as a whole is however shifted against the lower lying state diagram to smaller values of detector energy. The magnitude of the shift is determined by the energy separation of lower lying and excited state. The overlay of the two transfer diagrams then strongly depends on cross correlations between the two interacting detector levels. It turns out that an adequate treatment of these cross correlations requires us to develop further and advanced methodological tools, which is now on our agenda for a future project.

REVIEWERS' COMMENTS:

Reviewer #1 (Remarks to the Author):

The authors have revised the manuscript. All of the revisions are reasonable, and the main points of this paper become more convincing.

I recommend the paper for publication in Nature Communications.

The following is a minor comment just for the authors, which will not affect the main points of this paper.

As to the electrical isolation of the sensor QD,

The authors mentioned that the vertical line in Extended Data Figure 6 at $V_{\text{Sens}} \sim -0.693\text{V}$ shows the current through the Sensor QD caused by the experimentally unavoidable slight misalignment between μ_{L} and μ_{R} .

But if I understand correctly, the current through the Sensor QD will only make $I_{\text{Sens}} = -I_{\text{Right}}$ (The positive direction of the current is defined into the device.). Then, $I_{\text{Sens}} + I_{\text{Right}}$ will be 0.

There might be a missing charge problem...

Reviewer #2 (Remarks to the Author):

Dear Editors,

I apologise for my delayed review. I have been traveling.

I am satisfied with the author's responses and modifications of the manuscript, and I recommend it for publication in Nature communications

Regards

Reviewer #3 (Remarks to the Author):

The authors have adequately addressed my comments and questions.

I recommend publication without further modifications.

We thank all Reviewers for their comments and suggestions to improve the present manuscript entitled: "Auger-spectroscopy in quantum Hall edge channels: a possible resolution to the missing energy problem".

Below we provide answers to all of the reviewers' questions. These answers as well as the corresponding corrections are highlighted in the manuscript in red.

Reviewer #1

As to the electrical isolation of the sensor QD, The authors mentioned that the vertical line in Extended Data Figure 6 at $V_{\text{Sens}} \sim -0.693\text{V}$ shows the current through the Sensor QD caused by the experimentally unavoidable slight misalignment between μ_{L} and μ_{R} . But if I understand correctly, the current through the Sensor QD will only make $I_{\text{Sens}} = -I_{\text{Right}}$ (The positive direction of the current is defined into the device.). Then, $I_{\text{Sens}} + I_{\text{Right}}$ will be 0. There might be a missing charge problem...

Answer: The reasoning of Reviewer #1 is certainly correct and we mention the line is of similar origin, but it indeed cannot be explained by the misalignment of chemical potentials alone. At the point where the quantum dot chemical potential is aligned with the reservoirs the current through the device is very sensitive to noise fluctuations (on the gate voltages but also on the bias voltages). The voltage measurement of the IV-converter outputs happen sequentially for each pixel and thus the two signals do not correspond to the same noise fluctuations. Hence we can probe a difference in I_{Sens} to I_{Right} .